# Learning Re-sampling Methods with Parameter Attribution for Image Super-resolution

**Xiaotong Luo[1], Yuan Xie[2], Yanyun Qu[1]***

[1]School of Informatics, Xiamen University, Fujian, China
[2]School of Computer Science and Technology, East China Normal University, Shanghai, China
[2]Chongqing Institute of East China Normal University, Chongqing, China
`xiaotluo@stu.xmu.edu.cn, yxie@cs.ecnu.edu.cn, yyqu@xmu.edu.cn`

## Abstract

Single image super-resolution (SISR) has made a significant breakthrough benefiting from the prevalent rise of deep neural networks and large-scale training samples. The mainstream deep SR models primarily focus on network architecture design as well as optimization schemes, while few pay attention to the training data. In fact, most of the existing SR methods train the model on uniformly sampled patch pairs from the whole image. However, the uneven image content makes the training data present an unbalanced distribution, i.e., the easily reconstructed region (smooth) occupies the majority of the data, while the hard reconstructed region (edge or texture) has rarely few samples. Based on this phenomenon, we consider rethinking the current paradigm of merely using uniform data sampling way for training SR models. In this paper, we propose a simple yet effective Bi-Sampling Parameter Attribution (BSPA) method for accurate image SR. Specifically, the bi-sampling consists of uniform sampling and inverse sampling, which is introduced to reconcile the unbalanced inherent data bias. The former aims to keep the intrinsic data distribution, and the latter is designed to enhance the feature extraction ability of the model on the hard samples. Moreover, integrated gradient is introduced to attribute the contribution of each parameter in the alternate models trained by both sampling data so as to filter the trivial parameters for further dynamic refinement. By progressively decoupling the allocation of parameters, the SR model can learn a more compact representation. Extensive experiments on publicly available datasets demonstrate that our proposal can effectively boost the performance of baseline methods from the data re-sampling view.

## 1 Introduction

Single image super-resolution (SISR) aims to reconstruct a high-resolution (HR) image from its downsampled low-resolution (LR) counterpart. As a classical ill-posed problem, multiple HR images can be recovered from the same LR image. With the boom of neural networks, numerous excellent deep SR methods [9, 23, 25, 5] have been proposed to chase for a more accurate solution. It is well-known that the success of deep learning primarily attributes to three significant factors, including large-scale data [1], network structures [10, 52, 23], and optimization strategies [16, 31, 26]. Most of the existing SR models mainly focus on the delicate structure design or complex regularization constraints, while neglecting the in-depth analysis of the intrinsic training data.

Due to resource constraints, the mainstream deep SR methods mostly train the models with uniformly sampled LR-HR patch pairs rather than the whole images. However, they ignore the

---

*Corresponding author

underlying fact that the content of patches is usually unevenly distributed in an image. As presented in Fig. 1, we measure the patch content by mean square error (MSE) and count it for all the cropped sub-images on the commonly used DIV2K training dataset. It reveals a phenomenon of long-tailed distribution, i.e., the flat region occupies the majority of the training samples, while the sharp region with abundant texture details only holds a very small percentage.

The imbalanced training data would undoubtedly impair the reconstruction accuracy, especially for the hard areas in the tail. Therefore, we consider enhancing the model representation from the data sampling perspective.

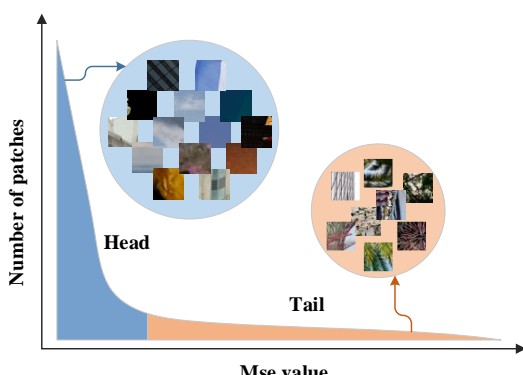

Figure 1: The illustration of long-tailed data distribution with uniform sampling in image SR. Following [19], we first crop the commonly used training dataset, e.g, DIV2K [1], into multiple sub-images, and adopt the mean square error (MSE) between the SR output by MSRResNet [40] and the HR image to measure the reconstruction difficulty of an LR sub-image. It shows that the data distribution of the training samples is uneven, which presents a long-tailed distribution.

In this paper, we propose a simple yet effective Bi-Sampling Parameter Attribution (BSPA) method for image SR, which aims to obtain the compact feature representation by explicitly increasing the sampling proportion of the tail hard patch pairs.

Specifically, the bi-sampling consists of uniform sampling and inverse sampling to make up for the unbalanced training data bias. As the most common way of data sampling, uniform sampling helps to keep the original data distributions, while inverse sampling aims to allocate a higher probability to the fewer patches sampled from the tail data. The SR model is alternately trained with the two kinds of sampling data. However, the SR models treat all the parameters equally and perform gradient updating to each parameter, which ignores the parameter redundancy. Therefore, it is necessary to obtain a compact model according to the parameter contribution to the output. Considering that the weight values would not change drastically with the small learning rate, it is difficult to determine the impact of different weights. Inspired by [35], we introduce the integrated gradient (IG) attribution method to measure the parameter importance. The IG is calculated between the previous model trained with uniform sampling data and the next model trained with inverse sampling data. Based on the importance ranking, the significant parameters do not perform gradient updating, and the remaining trivial parameters are encouraged to contribute more by further optimization. Extensive experiments have demonstrated that our proposal can improve the SR performance of baseline methods. To the best of our knowledge, this is the first attempt to investigate image SR as a long-tail data distribution problem.

In summary, the main contributions of this paper are: **1)** We definitely find the data unbalanced problem in the training samples caused by uniform sampling for image SR. **2)** A bi-sampling paradigm is designed to remedy the unbalanced data bias, including uniform sampling and inverse sampling, respectively. **3)** To obtain a compact representation, an integrated gradient based attribution method is introduced for measuring the parameter importance so that the significant parameters are progressively selected to stop gradient updating for better distilling the remaining trivial parameters.

## 2 Related Work

### 2.1 Deep Image SR Models

Nowadays, numerous SR models have been proposed with remarkable performance. The pioneering work, SRCNN [9], proposed a three-layer network for image SR. Then, FSRCNN [10] and ESPCN [32] introduced deconvolution and sub-pixel upsampling for mapping LR features to HR images, which could largely reduce computation complexity. VDSR [17] introduced residual learning to extract missing high-frequency information. DRRN [18] and DRCN [18] adopted recursive neural networks to reduce the model parameters while achieving excellent performance. RCAN [52] adopted a residual-in-residual network design and embedded the channel attention in each residual block to build an extremely deep network for image SR. DIN [21] designed a deep interleaved network to

combine the information at different states. Wang et. al. [37] learned a scale-arbitrary image SR network from scale-specific networks. LatticeNet [27] proposed a lattice block to adaptively combine pair-wise residual blocks. All these SR models have been trained on uniformly sampled LR-HR patch pairs, which ignore the unbalanced data distribution in the training samples.

Recently, several region-aware SR methods have been proposed to distinguish different image regions in an image and aim to assign more computation resources on more challenging pixels adaptively. RL-Restore [45] and Path-Restore [46] decomposed the image into different sub-images and then estimated an appropriate processing path for each sub-image by reinforcement learning. ClassSR [19] proposed a new classification method to determine the processing of different image regions and then reconstructed the sub-image by models of different sizes. Similar to ClassSR, FADN [43] proposed a framework for processing the different image regions divided by the corresponding information in the frequency domain with different computation burdens. APE [38] introduced an adaptive patch exiting strategy to super-resolve patches with different difficulties for achieving a flexible tradeoff between performance and efficiency. The above content-related SR methods mainly focus on the architecture design or optimization schemes. Except for the sub-image decomposition methods, UDN [31] proposed a new adaptive weighted loss for image SR to train deep networks focusing on challenging situations such as textured and edge pixels with high uncertainty.

In this paper, we mainly rethink the unbalanced distribution for image SR from the view of data sampling. SamplingAug [39] is a sampling-based data augmentation method for image SR. It only sampled the $p\%$ most informative patches for model training, while the rest is ignored. DDA [49] designed a difficulty-aware data sampling strategy, which controls the sampling probability of each class by the relative loss. Different from these works, we propose a bi-sampling strategy and the model is alternatively updated with uniformly and inversely sampled image data. All patches are effectively utilized for SR training to help the model fully mine the image information.

## 2.2 Long-tail Image Classification

Due to the imbalanced data distribution caused by the difficult data acquisition in actual applications, long-tailed image recognition has attracted widespread attention [41, 11, 50]. The existing long-tail studies can be roughly divided into three types: data re-sampling, loss re-weighting, and transfer learning. The data re-sampling based methods [28, 12] aim to re-sample the training dataset for achieving a more balanced distribution. Besides, the loss re-weighting based methods [13, 33] are proposed to allocate different weights to each training samples for loss optimization. Recently, the transfer learning based methods [44, 24] have sprung up to transfer the informative features learned from the head classes with abundant training samples to the under-represented tail classes. Here, we mainly explore the data re-sampling method for image SR.

As the widely adopted data re-balancing strategy for solving the long-tailed problem, re-sampling methods aim to sample the data to generate an evenly distributed dataset, which can be classified as: over-sampling for minority classes and under-sampling for dominant classes. However, repeated sampling for tailed samples may lead to the over-fitting problem on minority classes, while discarding the head samples will undoubtedly damage the generalization ability of deep models. Therefore, we integrate the two data sampling ways together for remedying the dataset bias in image SR task.

## 2.3 Attribution Analysis

Attribution analysis is usually utilized for model interpretation on the high-level image classification task, which aims to attribute the influence of each pixel in the input on prediction accuracy. The existing attribution analysis methods can be classified as: the gradient-based methods [34, 35], propagation-based methods [30, 4] and occlusion-based methods [47, 22]. Here, we primarily focus on the gradient-based methods, which utilize the gradient of the output against the input as a metric and accumulate the integrated gradients (IG) at all points along a path from the baseline input to the target input. IntInf [20] and neuron conductance [8] further extend feature-important IG to neurons. Unlike the classical attribution methods [35, 36], which are used to attribute the influence of the input on the classification model by attributing the contribution of each pixel of the input image to the final output label, FAIG [42] first introduced IG to attribute network functions to the filters in the blind SR model and successfully find discriminative filters that closely relate to the image degradation removal. In this paper, we mainly adopt the integrated gradient method to help locate the position of important weights in SR models trained with uniform sampling data and inverse sampling data.

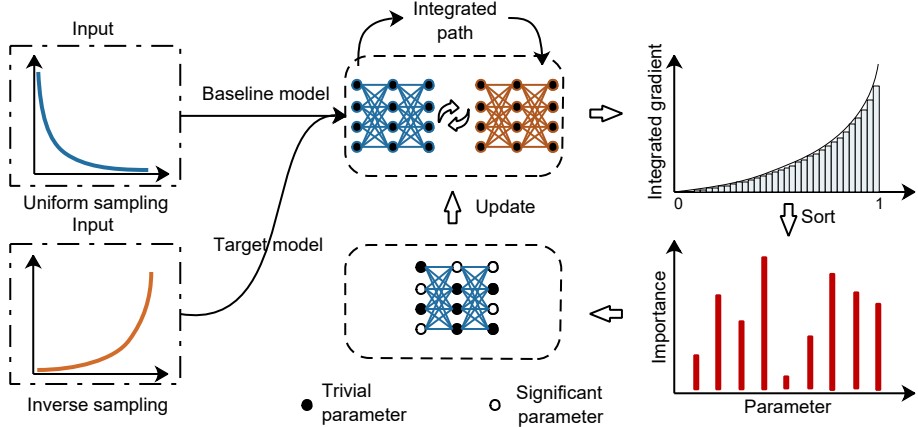

Figure 2: The framework of our proposed bi-sampling parameter attribution method for image SR.

## 3 Proposed Method

### 3.1 Method Overview

Considering the intrinsic dataset-specific bias shown in Fig. 1, we adopt two kinds of sampling ways (bi-sampling) for capturing the training data: uniform sampling and inverse sampling, which are combined to re-balance the distribution of training data. As shown in Fig 2, the proposed bi-sampling parameter attribution (BSPA) framework includes two steps, which are alternately performed. In Step 1, the SR model is trained with the data from uniform sampling to obtain a trivial solution toward normal data distribution (baseline model). In Step 2, the SR model is fine-tuned with the data from inverse sampling for generating a biased solution towards processing well on the tail edge and texture regions (target model). After training with inverse sampling data, the integrated gradient is calculated between the baseline model and the target model, which is utilized to obtain the parameter importance so as to control the relevant weight updating.

### 3.2 Bi-sampling Strategy

**Uniform Sampling.** Uniform sampling refers to cropping the patch samples from the whole image in the training dataset randomly, which is the most common way of data sampling and widely used in image SR task [21]. To be specific, the input data comes from a uniform sampler, where each sample in the training dataset is sampled with equal probability in a training epoch. The probability $P_i^{us}$ that a sample $i$ is chosen from the whole training dataset can be formulated as:

$$P_i^{us} = \frac{1}{N_T},\tag{1}$$

where $N_T$ is the total number of training samples.

**Inverse Sampling.** Inverse sampling aims to allocate a higher probability to capture the tail hard data with few samples. Unlike long-tailed image classification, which has a definite number of classes, image SR is a regression task. We adopt MSE to measure the reconstruction difficulties, which is a continuous value and each value corresponds to a sample as shown in Fig. 1. Considering the MSE value is continuous and divergently distributed, it needs to collect more samples within a certain range to calculate the proportion. Therefore, we first split data patches into multiple groups to better calculate the sampling probability.

*Data classification.* Specifically, the number of classes is predefined as $K$. Analogy to the long-tailed CIFAR, which is created by reducing the number of training samples per class via an exponential function [51], then the number of samples for each class can be calculated as:

$$N_k = N \times \mu^k,\tag{2}$$

where $k \in (1, K)$ is the class index, $N$ is the total number of training sub-images, and $\mu \in (0, 1)$ is an attenuation factor. Therefore, a long-tailed version of DIV2K training dataset can be created by setting different MSE threshold values.

Moreover, the imbalance factor of a long-tailed dataset is defined as the number of training samples in the largest class ($N_1$) divided by that of the smallest ($N_K$), which usually ranges from 10 to 200 in long-tail research and is set as 10 in our experimental setting. Finally, the MSE threshold values used for classifying the sub-images can be obtained by:

$$t_k = t[N_{1:k}], \tag{3}$$

where $t_k$ is the MSE threshold sorted by ascending. $N_{1:k}$ refers to the total number of sample from $N_1$ to $N_k$. Therefore, we can obtain the long-tailed version of DIV2K training dataset. It is noted that the test dataset remains unchanged and the SR model is still tested on the whole image.

*Sampling procedure.* Following [53], the sampling possibility of each class for inverse sampling $P_k^{is}$ is inversely proportional to the class sample capacity, i.e., the more samples in a class, the small sampling possibility that class has. Therefore, the procedures of inverse sampling are as follows: **1)** $P_k^{is}$ for the class $k$ can be calculated according to the number of samples as:

$$P_k^{is} = \frac{\frac{1}{N_k}}{\sum_{j=1}^{K} \frac{1}{N_j}}. \tag{4}$$

**2)** sample a class $k$ according to $P_k^{is}$ randomly; **3)** select a sample from class $k$ randomly. The mini-batch inverse sampling data can be obtained by repeating the above operations.

### 3.3 Integrated Gradient for Parameter Attribution

**Parameter Importance Formulation.** The uniform sampling data is adopted to maintain the inherent data distribution. The inverse sampling data is used to train the SR model for providing an elegant solution on edge and texture areas. In this paper, we adopt a hybrid training mechanism with uniform and inverse sampling data, which are iteratively input into the SR model.

Since the SR model is trained on uniform sampling and inverse sampling iteratively, there exist different preferences in the SR reconstruction regions. Then, it is meaningful to calculate the parameter importance for obtaining a compact model. Assume the parameter sets of the SR model $F(\cdot)$ trained with uniform sampling data and inverse sampling data are denoted as $\{\theta_i^{us} : 1 \leq i \leq L\}$ and $\{\theta_i^{is} : 1 \leq i \leq L\}$, where $\theta_i^{us}, \theta_i^{is} \in \mathbb{R}^{c_o \times c_i \times w \times h}$ are the weight matrixs at the $i$-th layer, $L$ is the number of convolutional layers, $c_o$ and $c_i$ are the number of output and input channels, $w$ and $h$ are the kernel sizes. The two sampling LR images are denoted as $X^{us}, X^{is}$, and the corresponding HR images are $Y^{us}, Y^{is}$, respectively. Then, for a uniformly sampled patch pair $(x^{us}, y^{us})$, the variation of network performance on the baseline model and the target model can be measured by:

$$\Delta F(\theta^{is}, \theta^{us}; x^{us}, y^{us}) = \left| l(F(x^{us}; \theta^{is}), y^{us}) - l(F(x^{us}; \theta^{us}), y^{us}) \right|, \tag{5}$$

where $l$ means the loss function used for training the SR network. For simplicity, we denote $l(F(x; \theta), y)$ as $l(\theta; x, y)$.

Take the element of parameter $\theta_{ik}^{us}$ ($i$-th layer) for example, the change of parameter $\theta_{ik}^{us}$ when the model is trained on the inverse sampling data is represented as $\delta\theta_{ik}^{us}$, i.e.,

$$\delta\theta_{ik}^{us} = \theta_{ik}^{is} - \theta_{ik}^{us}, \tag{6}$$

To further measure $\Delta F(\theta^{is}, \theta^{us}; x^{us}, y^{us})$, Cauchy mean value theorem [7] demonstates that when a function $f$ is differentiable at point $x_0$, it always stands up in the neighborhood $U(\cdot)$ of the point $x_0$:

$$f(x) = f(x_0) + f'(x_0)(x - x_0) + o(x - x_0), \tag{7}$$

Since $o(x - x_0)$ is an infinitesimal, it can be derived that $f(x) \approx f(x_0) + f'(x_0)(x - x_0)$. This is one of the formulas that is often used in local linearization of a function. Geometrically, it adopts the tangent line to approximate the curve in a small neighborhood.

Then, Eq. (5) can be further formulated as:

$$l\left(\theta_{ik}^{us} + \delta\theta_{ik}^{us}; x^{us}, y^{us}\right) - l\left(\theta_{ik}^{us}; x^{us}, y^{us}\right) \approx \left(\nabla_{\theta_{ik}^{us}} l\right)^T \cdot \delta\theta_{ik}^{us}. \tag{8}$$

**Integrated Gradient.** The above formula means that we can calculate the parameter importance by the value change and its gradient, which both consider the influence of network alteration on the model outputs. Here, we apply the integrated gradients (IG) to calculate the parameter importance, which was originally an attribution method to attribute the prediction of a deep network to its input features by accumulating the gradient of input changes. Inspired by FAIG [42], which utilizes paths in the parameter space for attributing network functional alterations to filter changes, we adopt IG to

**Algorithm 1** The bi-sampling parameter attribution for compact image SR.

---

**Input**: the SR model $F(\cdot)$ with the randomly initialized weights: $\theta_0$; the uniform sampling and inverse sampling datasets: $\boldsymbol{D^{us}}$, $\boldsymbol{D^{is}}$; the total epochs: $T$; the interval for training $F(\cdot)$ with inverse sampling: $E$; the remaining ratio for parameters without gradient updating: $p$.
**Output**: the compact SR model $F(\cdot)$ with parameter $\theta$.

 1: **while** *epoch* $\leq T$ **do**
 2:     Sample the LR-HR patch pairs $(x_{lr}^{us}, y_{hr}^{us})$ from $\boldsymbol{D^{us}}$;
 3:     Update $F(\cdot)$ with $\theta^{us}$ on the uniformly sampled data;
 4:     **if** *epoch* % $E$ = 0 **then**
 5:         Sample the LR-HR patch pairs $(x_{lr}^{is}, y_{hr}^{is})$ from $\boldsymbol{D^{is}}$;
 6:         Update $F(\cdot)$ with $\theta^{is}$ on the inversely sampled data;
 7:         Calculate the integrated gradient between the baseline model ($\theta^{us}$) and the target model ($\theta^{is}$) as Eq. (10) to obtain the parameter importance;
 8:         Keep $p$ significant parameters stop gradient updating as Eq.(11) after sorting the importance;
 9:     **end if**
10: **end while**

---

obtain the parameter importance so as to help us discover the location of significant parameters in the sampling iteration.

The integrated path is calculated from the model trained with uniform sampling data to the model trained with inverse sampling data. Let $\gamma(\alpha), \alpha \in [0, 1]$ denote a continuous path between the baseline model ($\theta'$) and the target model ($\theta$) with $\gamma(0) = \theta', \gamma(1) = \theta$. Then, the integrated path can be represented as:

$$\gamma(\alpha) = \theta' + \alpha(\theta - \theta'). \tag{9}$$

In addition, the $L_1$ loss is utilized to train the SR model, while the gradient of each parameter can be obtained according to the back-propagation algorithm. After defining the baseline and target model, integarted path and the loss function, the changes of network functions to each parameter with integrated gradient for an input image $x$ can be attributed as follows:

$$\begin{aligned}
IG(\theta, x) &= l(\gamma(1), x) - l(\gamma(0), x) \\
&= \sum_i \int_{\alpha=0}^{1} \frac{\partial l(\gamma(\alpha), x)}{\partial \gamma(\alpha)_i} \times \frac{\partial \gamma(\alpha)_i}{\partial \alpha} \mathrm{d}\alpha \\
&\approx \frac{1}{M} [\theta - \theta'] \sum_{s=0}^{M-1} \left[ \frac{\partial l(\gamma(\alpha), x)}{\partial \gamma(\alpha)} |_{\gamma(\alpha) = \theta' + \alpha(\theta - \theta'), \alpha = \frac{s}{M}} \right].
\end{aligned} \tag{10}$$

**Parameter Refinement.** After calculating the integrated gradient, the position index of parameter importance can be located. The alternate training scheme is adopted for the bi-sampling data. To obtain a more compact representation, we expect the trivial parameters can also learn more effective information so we strive to distill them with other parameters fixed. Therefore, a specific proportion of significant parameters can be selected to keep unchanged and only perform the gradient updating on the remaining trivial parameters for further refinement. Especially, the selective ratio is set as:

$$p = \beta * (e/T), \tag{11}$$

where $e$ is the $e$-th training epoch, and $T$ is the total training epochs. $\beta$ is a scaling factor. The detailed training procedure of our proposal is given in Algorithm 1.

## 4 Experiments

### 4.1 Datasets and Implementation Details

**Datasets**. We use DIV2K [1] to train the SR models, which is a high-quality dataset widely used for image SR. The whole dataset includes 800 training images and 100 validation images totally with diverse contents and texture details. The LR images are obtained in the same way as [52, 15]. To demonstrate the effectiveness of our method, the SR models are also evaluated on five public SR benchmark datasets: Set5 [3], Set14 [48], B100 [2], Urban100 [14] and Manga109 [29].

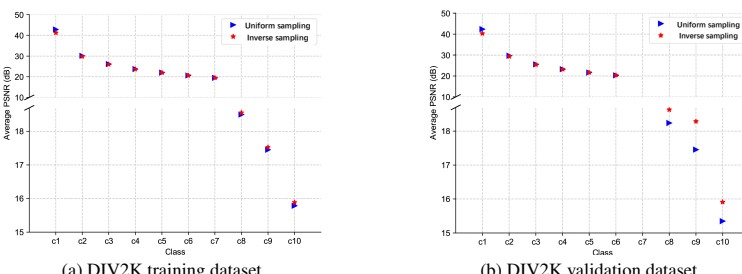

(a) DIV2K training dataset          (b) DIV2K validation dataset

Figure 3: The PSNR of each class on the classified DIV2K training and validation datasets for uniform sampling and inverse sampling.

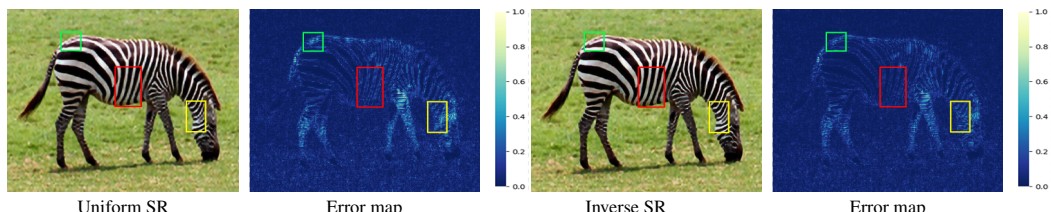

Uniform SR       Error map       Inverse SR       Error map

Figure 4: Visual comparisons of the SR results by the uniform sampling (uniform SR) and the inverse sampling (inverse SR), and the corresponding error maps with the GT.

**Implementation Details**. During training, we fix the patch size of the HR image as $128 \times 128$ for $2\times$, $4\times$ SR, and $129 \times 129$ for $3\times$ SR. We follow ClassSR [19] for data preprocessing in inverse sampling, which crops the whole image into multiple sub-images with sliding and overlap. For $2\times$, $3\times$ SR, $4\times$ SR, it produces $499875$, $502200$, $499875$ sub-images, respectively. Following [39], the number of the data repetition is set to $1$. The training data is augmented by random flipping and rotation. We use Adam optimizer with $\beta_1 = 0.9$, $\beta_2 = 0.999$ to train the SR models. The mini-batch size is set to $16$. The learning rate is initialized as $2e - 4$ and reduced by half per $200$ epochs for $400$ epochs totally. The unbalanced factor for the inverse sampling data is set to $10$ and $\beta$ is set to $0.1$. The interval of alternate training is $50$ epochs and the number of classes of inverse sampling for DIV2K training dataset is $10$. In our experiments, $\mu$ is derived to $0.7743$ according to the number of classes and the unbalanced factor. All the experiments are conducted with PyTorch framework on NVIDIA 2080Ti GPUs. Besides, objective criteria, i.e., peak signal-to-noise ratio (PSNR), structural similarity (SSIM) are utilized to evaluate the model performance. Note that the two metrics are both calculated on the Y channel of the YCbCr space as adopted in the previous works [6, 52].

## 4.2 Ablation Study

In this subsection, we mainly give a detailed analysis about the proposed bi-sampling parameter attribution (BSPA) for accurate image SR. Following [31], EDSR_baseline (16 residual blocks with 64 feature channels) is exploited as the backbone network.

**Bi-sampling Strategy.** The bi-sampling strategy is adopted to capture the training samples for the two-step iterative optimization procedure. Uniform sampling ensures the original data distribution, while inverse sampling mainly enhances the feature extraction for the edge and texture regions. Note that all the cropped sub-images are grouped as 10 classes for inverse sampling in our experiments. In Tab. 1, it is observed that the baseline model obtains 28.22 dB in PSNR on Set14, while the model trained with inverse sampling for unbalanced factor 10 achieves 28.25 dB. Besides, we present the average PSNR value of each class on the cropped DIV2K training and validation sub-image dataset for the SR model separately trained by uniform sampling and inverse sampling in Fig. 3. It illustrates that the

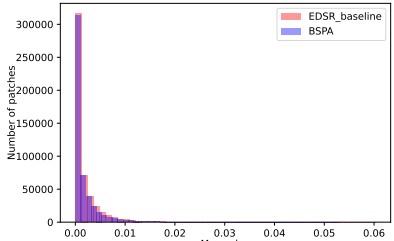

Figure 5: The histogram distribution of all training sub-images in DIV2K for EDSR_baseline and BSPA ($4 \times SR$). It is observed that the distribution of BSPA becomes short-tailed.

Table 1: Ablation studies about the bi-sampling parameter attribution on benchmarks for $4\times$ SR.

| Model | Baseline (uniform sampling) | inverse sampling | w/o IG | FIG | WIG |
|---|---|---|---|---|---|
| Set14 | 28.22 | 28.25 | 28.28 | 28.32 | 28.37 |
| B100 | 27.31 | 27.33 | 27.35 | 27.38 | 27.40 |
| Urban100 | 25.27 | 25.34 | 25.38 | 25.47 | 25.53 |
| DIV2K valid | 29.97 | 30.02 | 30.06 | 30.09 | 30.13 |

Table 2: The quantitative comparisons (PSNR) of different selective ratios on benchmarks for $4\times$ SR.

| $\beta$ | 0.1 | 0.5 | 0.8 | 1.0 |
|---|---|---|---|---|
| Set14 | 28.37 | 28.35 | 28.34 | 28.32 |
| B100 | 27.40 | 27.37 | 27.35 | 27.31 |
| Urban100 | 25.53 | 25.51 | 25.46 | 25.43 |
| DIV2K valid | 30.13 | 30.10 | 30.07 | 30.05 |

Table 3: The quantitative comparisons (PSNR) of different unbalanced factors on benchmarks for $4\times$ SR.

| Unbalanced factor | 10 | 50 | 100 | 200 |
|---|---|---|---|---|
| Set14 | 28.37 | 28.33 | 28.28 | 28.21 |
| B100 | 27.40 | 27.35 | 27.31 | 27.23 |
| Urban100 | 25.53 | 25.49 | 25.42 | 25.30 |
| DIV2K valid | 30.13 | 30.06 | 30.00 | 29.91 |

SR model trained with inverse sampling performs much better than the SR model with the uniform sampling on the tail hard classes, while performing worse than the SR model with the uniform sampling on the simple classes. Therefore, inverse sampling helps the feature extraction on hard texture regions. We also visualize the SR results and their error maps with the GT of uniform sampling (uniform SR) and inverse sampling (inverse SR) in Fig. 4. It shows that inverse SR has a smaller error than the uniform SR on the texture region.

**Integrated Gradient.** To demonstrate the effect of the integrated gradient (IG), we first use uniformly sampled and inversely sampled data to alternately train the SR model. As the model "w/o IG" shows in Tab. 1, the hybrid training obtains $0.06$ dB improvement in PSNR against the baseline model on Set14. Besides, we compare the filter-level IG (FIG) and weight-level IG (WIG), which denote the parameter granularity in Eq.10 for each filter and each parameter. It is observed that WIG achieves the best PSNR, which illustrates that the weight is more discriminative than the filter for the model performance. Note that we adopt the weight-level IG in the experiments. Besides, we visualize the distribution curve if the MSE values are evaluated by the model trained with BSPA. As Fig. 5 shows, it is observed that the histogram distribution of BSPA becomes short-tailed. The reason is that the SR performance on the tail data is improved, while the performance on the head data is kept. Therefore, it demonstrates that our BSPA is effective in obtaining a more compact representation.

**Effect of Selective Ratio.** The selective factor in Eq. (11) aims to control the proportion of the significant parameters and update the remaining trivial parameters. As shown in Tab. 2, we give the quantitative comparisons of different $\beta$ values for the selective ratio. It can be observed that the best PSNR is obtained when $\beta$ is $0.1$. Therefore, we fix $\beta$ as $0.1$ in our experiments.

**Effect of Unbalanced Factor.** To demonstrate the effect of the unbalanced factor used for inverse sampling data generation, we give the results of our BSPA with different unbalanced factors as shown in Table 3. It shows that the best PSNR is obtained when the unbalanced factor is equal to 10. Since the unbalanced factor determines the sampling probability of each class in inverse sampling, the large factor causes multiple repetitive sampling to the tail hard samples. It may bring about the overfitting on these areas so as to hinder the model representation.

**Effect of the Number of Classes.** For the classes, we compare different number of classes in Table 5. It shows that the SR performance changes little to the number of classes. Because the data in different classes can be sampled according to their proportions by means of inverse sampling so as to remedy the unbalanced data bias. In our experiments, we set the number of classes to 10.

## 4.3 Comparisons with the State-of-the-arts

To demonstrate the effectiveness of our BSPA, we integrate it with three representative SR models with different model capacities, i.e., FSRCNN [10] (small), EDSR_baseline [23] (medium), and RCAN [52] (large). Besides, we compare BSPA with SamplingAug [39] (a sampling-based data augmentation method) and UDN [31] (a loss reweighting method). All these SR models are retrained under the same experimental settings with the released codes.

**Quantitative Results.** In Tab. 4, it shows that SamplingAug and our BSPA outperform the baseline methods with different model capacities for all scaling factors, which is even higher $1.32$dB than the original FSRCNN on DIV2K validation dataset for $2\times$ SR. The results of UDN on some datasets are

Table 4: The quantitative comparison results of average PSNR/SSIM for $2\times$, $3\times$, $4\times$ image SR. The best results are highlighted in black bold and the second best is in underline.

| Scale | Method | Params | Set5 PSNR/SSIM | Set14 PSNR/SSIM | B100 PSNR/SSIM | Urban100 PSNR/SSIM | Manga109 PSNR/SSIM | DIV2K valid PSNR/SSIM |
|---|---|---|---|---|---|---|---|---|
| $2\times$ | FSRCNN [10] | 24.5K | 33.09/0.9180 | 30.02/0.8625 | 29.44/0.8396 | 26.77/0.8320 | 30.29/0.9172 | 32.18/0.8989 |
|  | SamplingAug [39] | 24.5K | 34.71/0.9414 | 31.24/0.8942 | 30.51/0.8766 | 27.83/0.8667 | 32.28/0.9452 | 33.46/0.9225 |
|  | UDN [31] | 24.5K | 34.79/0.9412 | 31.29/0.8940 | 30.50/0.8758 | 27.81/0.8660 | 32.34/0.9448 | 33.47/0.9223 |
|  | BSPA (ours) | 24.5K | 34.80/0.9420 | 31.27/0.8943 | 30.54/0.8772 | 27.86/0.8682 | 32.27/0.9454 | 33.50/0.9232 |
|  | EDSR_baseline [23] | 1.4M | 37.58/0.9590 | 33.15/0.9140 | 31.89/0.8959 | 30.89/0.9156 | 37.57/0.9745 | 35.54/0.9411 |
|  | SamplingAug [39] | 1.4M | 37.65/0.9591 | 33.25/0.9146 | 31.96/0.8969 | 31.15/0.9184 | 37.76/0.9747 | 35.66/0.9420 |
|  | UDN [31] | 1.4M | 37.60/0.9590 | 33.12/0.9137 | 31.89/0.8960 | 30.93/0.9160 | 37.67/0.9747 | 35.56/0.9411 |
|  | BSPA (ours) | 1.4M | 37.70/0.9594 | 33.27/0.9149 | 31.96/0.8968 | 31.18/0.9185 | 37.78/0.9752 | 35.69/0.9420 |
|  | RCAN [52] | 15.4M | 37.71/0.9596 | 33.31/0.9151 | 31.99/0.8976 | 31.41/0.9213 | 37.93/0.9755 | 35.74/0.9428 |
|  | SamplingAug [39] | 15.4M | 37.89/0.9599 | 33.47/0.9161 | 32.05/0.8978 | 31.69/0.9240 | 38.39/0.9762 | 35.91/0.9436 |
|  | UDN [31] | 15.4M | 37.86/0.9601 | 33.43/0.9162 | 32.05/0.8983 | 31.65/0.9239 | 38.15/0.9760 | 35.89/0.9440 |
|  | BSPA (ours) | 15.4M | 37.94/0.9603 | 33.58/0.9181 | 32.14/0.8994 | 31.98/0.9272 | 38.49/0.9767 | 36.06/0.9449 |
| $3\times$ | FSRCNN [10] | 24.5K | 29.91/0.8520 | 27.26/0.7625 | 27.07/0.7309 | 24.34/0.7237 | 26.46/0.8318 | 29.38/0.8221 |
|  | SamplingAug [39] | 24.5K | 31.02/0.8794 | 28.13/0.7978 | 27.69/0.7657 | 24.97/0.7574 | 27.58/0.8632 | 30.18/0.8476 |
|  | UDN [31] | 24.5K | 31.21/0.8825 | 28.27/0.8018 | 27.75/0.7701 | 25.03/0.7613 | 27.78/0.8682 | 30.28/0.8504 |
|  | BSPA (ours) | 24.5K | 31.13/0.8823 | 28.20/0.8008 | 27.74/0.7693 | 25.01/0.7611 | 27.65/0.8680 | 30.24/0.8498 |
|  | EDSR_baseline [23] | 1.6M | 33.79/0.9219 | 29.91/0.8340 | 28.83/0.7979 | 27.21/0.8305 | 32.30/0.9345 | 31.87/0.8787 |
|  | SamplingAug [39] | 1.6M | 33.85/0.9223 | 30.00/0.8353 | 28.86/0.7987 | 27.35/0.8344 | 32.55/0.9361 | 31.95/0.8800 |
|  | UDN [31] | 1.6M | 33.75/0.9216 | 29.89/0.8335 | 28.82/0.7978 | 27.15/0.8298 | 32.33/0.9344 | 31.85/0.8786 |
|  | BSPA (ours) | 1.6M | 33.97/0.9237 | 30.03/0.8361 | 28.89/0.7999 | 27.42/0.8359 | 32.60/0.9372 | 31.99/0.8807 |
|  | RCAN [52] | 15.6M | 33.70/0.9206 | 29.87/0.8338 | 28.78/0.7970 | 27.11/0.8274 | 32.10/0.9320 | 31.81/0.8776 |
|  | SamplingAug [39] | 15.6M | 34.15/0.9247 | 30.20/0.8390 | 28.96/0.8011 | 27.78/0.8439 | 33.13/0.9403 | 32.14/0.8830 |
|  | UDN [31] | 15.6M | 34.26/0.9255 | 30.22/0.8397 | 29.00/0.8026 | 27.80/0.8448 | 33.05/0.9407 | 32.23/0.8846 |
|  | BSPA (ours) | 15.6M | 34.35/0.9268 | 30.31/0.8413 | 29.07/0.8044 | 28.15/0.8519 | 33.35/0.9432 | 32.36/0.8867 |
| $4\times$ | FSRCNN [10] | 24.5K | 28.14/0.7960 | 25.91/0.6660 | 25.94/0.6660 | 23.14/0.6537 | 24.71/0.7705 | 28.00/0.7710 |
|  | SamplingAug [39] | 24.5K | 29.02/0.8242 | 26.52/0.7266 | 26.37/0.6936 | 23.59/0.6813 | 25.40/0.7950 | 28.57/0.7919 |
|  | UDN [31] | 24.5K | 29.11/0.8250 | 26.53/0.7272 | 26.37/0.6936 | 23.58/0.6811 | 25.45/0.7965 | 28.60/0.7927 |
|  | BSPA (ours) | 24.5K | 29.04/0.8241 | 26.51/0.7261 | 26.35/0.6921 | 23.55/0.6790 | 25.40/0.7957 | 28.56/0.7912 |
|  | EDSR_baseline [23] | 1.5M | 31.57/0.8861 | 28.22/0.7723 | 27.31/0.7268 | 25.27/0.7574 | 29.27/0.8902 | 29.97/0.8265 |
|  | SamplingAug [39] | 1.5M | 31.72/0.8876 | 28.29/0.7735 | 27.34/0.7281 | 25.39/0.7624 | 29.57/0.8936 | 30.04/0.8279 |
|  | UDN [31] | 1.5M | 31.44/0.8833 | 28.15/0.7707 | 27.27/0.7257 | 25.17/0.7539 | 29.05/0.8866 | 29.90/0.8253 |
|  | BSPA (ours) | 1.5M | 31.80/0.8897 | 28.37/0.7765 | 27.40/0.7312 | 25.53/0.7685 | 29.72/0.8972 | 30.13/0.8312 |
|  | RCAN [52] | 15.6M | 31.92/0.8910 | 28.38/0.7763 | 27.40/0.7306 | 25.54/0.7684 | 29.77/0.8982 | 30.16/0.8311 |
|  | SamplingAug [39] | 15.6M | 32.00/0.8915 | 28.48/0.7788 | 27.47/0.7323 | 25.80/0.7771 | 30.17/0.9027 | 30.23/0.8331 |
|  | UDN [31] | 15.6M | 31.91/0.8907 | 28.45/0.7773 | 27.45/0.7311 | 25.70/0.7721 | 29.86/0.8989 | 30.21/0.8321 |
|  | BSPA (ours) | 15.6M | 32.18/0.8943 | 28.56/0.7807 | 27.54/0.7348 | 26.02/0.7839 | 30.34/0.9060 | 30.39/0.8363 |

even inferior to the baseline model, which may originate from the inaccurate estimation of uncertainty. Note that the model's performance on FSRCNN for $3\times$ and $4\times$ scale is worse than UDN. FSRCNN is an extremely lightweight model, which has only $24.5$K parameters and several convolutional layers. The model representation ability is restricted. Therefore, it may not fit the training well no matter the easy data or the hard data by the limited model capacity, especially for large scaling factors. Although the bi-sampling strategy increases the data diversity, there lacks sufficient model capacity to it. UDN is a loss re-weighting method for solving the data imbalance problem. It provides a regularization constraint for shrinking the solution space. Besides, our method has advantages over other compared methods on large models. For EDSR and RCAN, our BSPA almost performs the best for all scaling factors against the compared methods.

**Qualitative Results.** The visual comparisons on benchmark datasets for $4\times$ SR are shown in Fig. 6. For images "img042" and "img092" in Urban100 dataset, we can observe that BSPA is more favorable and can recover more texture details than other compared methods. Therefore, our proposal can not only achieve excellent performance but also be superior in the visual effect. Besides, we also provide the perceptual metrics (NIQE/LPIPS) comparisons for RCAN in Table 6. It shows that our method obtains superior NIQE and LPIPS values against the compared methods.

Table 5: The quantitative comparisons (PSNR) of different numbers of classes on benchmarks for $4\times$ SR.

| Classes (K) | 5 | 10 | 20 |
|---|---|---|---|
| Set14 | 28.36 | 28.37 | 28.38 |
| B100 | 27.41 | 27.40 | 27.42 |
| Urban100 | 25.53 | 25.53 | 25.54 |
| DIV2K valid | 30.13 | 30.13 | 30.14 |

Table 6: The quantitative comparison results (NIQE/LPIPS) on benchmarks for $4\times$ SR.

| Model | Set14 | B100 | Urban100 | DIV2K valid |
|---|---|---|---|---|
| RCAN | 6.59/0.2932 | 6.73/0.3868 | 5.83/0.2508 | 6.03/0.2836 |
| SamplingAug | 6.33/0.2876 | 6.53/0.3809 | 5.60/0.2416 | 5.83/0.2784 |
| UDN | 6.61/0.2926 | 6.69/0.3850 | 5.76/0.2475 | 5.85/0.2813 |
| BSPA (ours) | 6.26/0.2844 | 6.46/0.3750 | 5.61/0.2314 | 5.79/0.2748 |

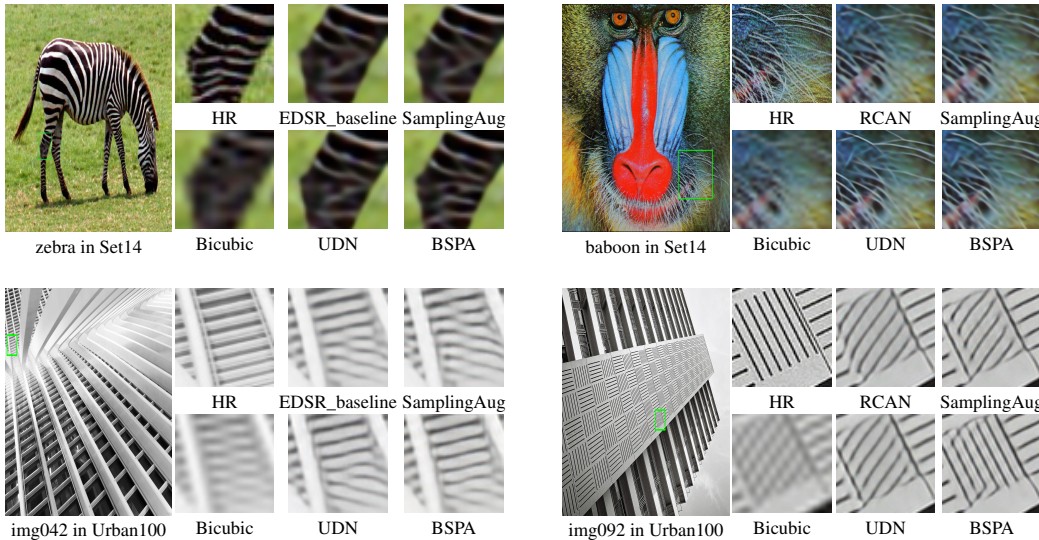

Figure 6: Visual comparisons of our BSPA with other methods on Set14 and Urban100 for $4\times$ SR.

## 4.4 Limitations

We adopt a re-sampling strategy to address the unbalanced data distribution problem for image SR. It has several limitations. **1)** Unlike the end-to-end loss re-weighting methods [31], our method needs to implement extra data pre-processing for classifying sub-images at first. **2)** Inverse sampling allocates larger weights for the tail training samples, which may lead to over-fit upon these data. Therefore, it is important to define an appropriate unbalanced factor. Most of the existing SR models only consider that different image regions have different reconstruction diîculties, and then mainly focus on the delicate structure design (like attention mechanism) or complex regularization constraints. Few discuss the sampling strategies of training data and model this as a long-tail problem. The key idea behind the proposed Bi-Sampling Parameter Attribution method is to reconcile the unbalanced inherent data bias, namely the heavy-tailed distribution are visually more important than smooth areas. The newly developed technique with integrate gradient involves an inverse sampling strategy for enhancing the feature extraction ability of the model on the hard samples. We believe that our method has such potential for fully mining the dataset information and may inspire the research of this direction in the future.

## 5 Conclusion

In this paper, we rethink the current paradigm of merely using uniform sampling to capture the mini-batch data for training the SR model, which would lead to the notorious data unbalanced problem. To address this issue, we propose a bi-sampling parameter attribution (BSPA) method for accurate image SR. The bi-sampling is introduced to remedy the unbalanced data distribution, including the usually adopted uniform sampling and inverse sampling. The former aims to keep the intrinsic data distribution, the latter is designed to promote the model performance on the few hard samples, which is inversely proportional to the number of each class. To achieve this, we divide the whole training dataset into multiple sub-images and group these sub-images into different classes according to the reconstruction error. Moreover, the integrated gradient method is introduced to calculate the parameter importance, with which the significant parameters are fixed and the remaining trivial parameters are further refined. Extensive experiments on SR benchmark datasets have demonstrated that our proposal achieves superior performance against the baseline methods. In our future work, we will apply the proposal to other low-level tasks based on patch training.

**Acknowledgments.** This work is supported by the National Key Research and Development Program of China No.2020AAA0108301; National Natural Science Foundation of China under Grants No.62176224 and No.62222602; Natural Science Foundation of Chongqing under No.CSTB2023NSCO-JOX0007; CCF-Lenovo Blue Ocean Research Fund.

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
