# OpenReview forum: "Learning Re-sampling Methods with Parameter Attribution for Image Super-resolution"
_NeurIPS.cc/2023/Conference — NeurIPS 2023 poster_

### Official Review · Reviewer_ehLM · 2023-06-24

**Soundness:** 3 good
**Presentation:** 3 good
**Contribution:** 3 good
**Rating:** 5
**Confidence:** 4

**Summary:**

The key idea behind the proposed Bi-Sampling 12 Parameter Attribution (BSPA) method is to reconcile the unbalanced inherent data bias, namely the heavy-tailed distribution are visually more important than smooth areas. Similar observation has been extensively observed in the literature of image restoration and super-resolution. The newly developed technique involves an inverse sampling strategy for enhancing the feature extraction ability of the model on the hard samples. Another technical contribution is to introduce integrated gradient for further improvement. Limited experimental results are reported to show marginal improvements over UDR (NeurIPS'2021).

**Strengths:**

1. The motivation behind BSPA is clear and well explained. Although similar observation exists in the literature of SR, sampling-based approach for unbalanced data has not been considered for SR before (previous studies on long-tailed data distribution are mostly for high-level vision tasks such as image recognition).
2. The proposed integrate gradient (IG) method is based on importance ranking and only updates the class of trivial parameters. This strategy is conceptually similar to the reweighting idea in the literature but easier to implement.
3. The reported experimental results have shown descent improvement over previous benchmark method such as UDN [27].

**Weaknesses:**

1. The proposed bi-sampling framework in Fig. 2 seems to be based on heuristics. When only LR observation is available, it is unclear from the figure how to adapt the second step toward the prioritization of edge and texture regions.
2. For inverse sampling, the procedure described in Eq. (4) lacks substantial justification. P(x)~1/x in Eq. (1) is difficult to follow (ref. [20] does not seem to directly suggest this formula).
3. The idea of integrated gradient is inherited from previous work FAIG [38]. What is the novel contribution here? It seems that the derivation of Eqs. (9)-(10) is a direct consequence of Eq. (4) in the original FAIG paper.
4. The reported experimental results do not support the claim of "significantly boost the performance" in the abstract. Both subjective and objective evaluation suggest that the performance of BSPA is comparable to that of other competing methods. With only one figure with small size images included, the superiority of BSPA is unconvincing.

**Questions:**

1. What is the point for Fig. 4 to make? It is not easy to see the difference between uniform and inverse SR in the current comparison (including both images and error maps).
2. Fig. 3 (a) looks confusing. Not much difference between random sampling and inverse sampling, what does this figure attempt to show?
3. The idea of "better distilling the remaining trivial parameters" is counter-intuitive. This could be an issue of literary presentation. If some parameters are not important (i.e., "trivial"), why do we strive to distill them better?
4. Why do you coin the term "bi-sampling"? Does uniform sampling and inverse sampling carry equal importance? If not, you might want to consider a more appropriate title for this work. Bi-sampling is different from bilateral filtering where the domain and range can be viewed as a dual representation. I am not sure if a similar duality holds for sampling procedure.

**Limitations:**

Have the authors considered other recent competing approaches to SR such as diffusion model-based?  The baseline methods used in this paper do not seem to represent the current SOTA in SR.

---

> ### Author Rebuttal · Authors · 2023-08-10
>
> Response to Reviewer xmJz (denoted as R5)
>
> *Q5-1: The proposed bi-sampling framework in Fig. 2 seems to be based on heuristics.*
>
> A5-1:  It is **unreasonable** to say the proposed framework is heuristics. We aim to propose a simple yet effective bi-sampling parameter attribution method for accurate image SR from the data re-sampling view. Similar to most of the existing SR works, our proposal is a **supervised** method and the LR-HR pairs are available. The bi-sampling strategy takes full advantage of the paired samples, where the inverse sampling captures more hard samples to train the model for performing well on edge and texture regions. The unsupervised case is out of our scope.
>
> *Q5-2: For inverse sampling, the procedure described in Eq. (4) lacks substantial justification. P(x)~1/x in Eq. (1) is difficult to follow (ref. [20] does not seem to directly suggest this formula).*
>
> A5-2: The reviewer might misunderstand this. 1) Eq.(4) presents the sampling possibility of each class for inverse sampling, which is inversely proportional to the number of samples in each class. 2) Eq.(1) denotes the sampling probability is the same for each sample in uniform sampling. More generally, it refers to random sampling, which is widely used in the existing SR model, e.g., ref[20]. However, due to the uneven content distribution within the image, the uniform sampling would cause the data unbalanced problem, which motivates us to design such a bi-sampling parameter attribution method.
>
> *Q5-3: The idea of integrated gradient is inherited from previous work FAIG [38]. What is the novel contribution here? It seems that the derivation of Eqs. (9)-(10) is a direct consequence of Eq. (4) in the original FAIG paper.*
>
> A5-3: The reviewer might misunderstand this. We do not claim integrated gradient is our novel contribution.
>
> （1) The key idea behind the proposed BSPA is to reconcile the unbalanced inherent data bias, for enhancing the feature extraction ability of the model on the hard samples. Therefore, we need to find the significant parameters for different sampling ways to obtain a compact representation space.
>
> （2) To achieve this, we first formulate the parameter importance with Cauchy mean value theorem as Eq. (8). The integrated gradient merely serves as a tool for the attribution analysis by splitting the weight changes into a continuous path, which is usually used in numerical analysis and other level tasks.
>
> *Q5-4: The reported experimental results do not support the claim of "significantly boost the performance" in the abstract. Both subjective and objective evaluation suggest that the performance of BSPA is comparable to that of other competing methods. With only one figure with small size images included, the superiority of BSPA is unconvincing.*
>
> A5-4: The reviewer might misunderstand this. 1) In the abstract, we claim that our proposal can significantly boost the performance of **baseline methods** from the data re-sampling view. Besides, the quantitative and qualitative results also illustrate our proposal achieves superior or comparable performance against the compared methods. 2) For the subjective evaluation, we supplemented more visual results in **Figure 1 of the uploaded PDF**. It is observed that our proposal is more favorable and recovers more texture details than other compared methods. 3) For the objective evaluation, we also supplement more results on other datasets (please refer to **A1-1 of Reviewer zt8x**), which shows the effectiveness of our BSPA. We hope R5 can notice these results and analyses.
>
> *Q5-5: What is the point for Fig. 4 to make? It is not easy to see the difference between uniform and inverse SR in the current comparison (including both images and error maps).*
>
> A5-5: Fig.4 visualizes the SR results and their error maps with the GT of uniform sampling (uniform SR) and inverse sampling (inverse SR). It shows that inverse SR has a smaller error than the uniform SR on the texture region. Please zoom in for a better view, especially for the framed regions. Due to the limited space, we will provide more results in the new version.
>
> *Q5-6: Fig. 3 (a) looks confusing. Not much difference between random sampling and inverse sampling, what does this figure attempt to show?*
>
> A5-6: Fig.3 (a) illustrates that inverse sampling helps the feature extraction on hard texture regions, and performs better on tail hard classes against uniform sampling on the training dataset. The heavy-tailed distribution are visually more important than smooth areas, which is difficult for image SR. Since the scale of the ordinate is larger, you can zoom in for a better view.
>
> *Q5-7: The idea of "better distilling the remaining trivial parameters" is counter-intuitive. This could be an issue of literary
> presentation. If some parameters are not important (i.e., "trivial"), why do we strive to distill them better?*
>
> A5-7: The reviewer might misunderstand this. In 'parameter refinement', we select a specific proportion of significant parameters according to the index sorting to keep unchanged, and only perform the gradient updating on the remaining trivial parameters for further refinement. The alternate training scheme is designed for distilling all parameters to obtain a more compact representation space. We expect the trivial parameters can also learn more effective information so we strive to distill them with other parameters fixed.
>
> *Q5-8: Why do you coin the term "bi-sampling"? Does uniform sampling and inverse sampling carry equal importance? If not, you might want to consider a more appropriate title for this work. Bi-sampling is different from bilateral filtering where the domain and range can be viewed as a dual representation.*
>
> A5-8: The bi-sampling refers to the two kinds of sampling ways, i.e., uniform sampling and inverse sampling. In our proposal, uniform sampling and inverse sampling carry equal importance.

---

> > ### Author Response · Authors · 2023-08-19
> > **Further discussion**
> >
> > We hope to further discuss with you whether your concerns have been addressed or not. If you still have any unclear parts of our work, please let us know. Thanks.

---

> > ### Comment · Reviewer_ehLM · 2023-08-19
> > **Visual quality comparison**
> >
> > I am mostly satisfied with the authors' rebuttal. Therefore, I have increased my rating by one level. However, I was hoping that the authors could provide a more detailed comparison in terms of subjective quality. For image SR study, PSNR differences matter less than visual quality improvement, in my biased opinion. This paper only contains a small figure (Fig. 5) for subjective quality comparison. If the proposed BSPA framework works so well, it should be fairly easy to find challenging examples from the distribution tail and report significant improvement in terms of visual quality.

---

> > > ### Author Response · Authors · 2023-08-19
> > > **Thanks Reviewer ehLM for approving our response**
> > >
> > > Dear Reviewer ehLM,
> > >
> > > Thanks for agreeing that our response solves the concerns. Generally, the visual quality is measured by the **visual results** or the **perceptual metrics** like NIQE (a non-reference metric) and LPIPS (a reference-based metric that computes the perceptual similarity between the ground truth and the output SR image).
> > >
> > > For the visual results, we have supplemented more examples in **Figure 1 of the uploaded PDF in the front of this webpage**. It is observed that our proposal is more favorable and recovers more texture details (hard regions) than other compared methods.
> > >
> > > For the perceptual metrics, we provide the **NIQE/LPIPS** comparisons for RCAN (4x SR) in the following table. It shows that our method obtains superior NIQE and LPIPS values against the compared methods.
> > >
> > > |Model |	Set14 |	B100 | Urban100 | DIV2K_valid |
> > > |------ |:-----:|:-----:|:-----:|--------:|
> > > | RCAN | 6.59/0.2932 |	6.73/0.3868 |	5.83/0.2508 |	6.03/0.2836 |
> > > | SamplingAug  | 6.33/0.2876 |	6.53/0.3809 |	5.60/0.2416 |	5.83/0.2784 |
> > > | UDN | 6.61/0.2926 |	6.69/0.3850 |	5.76/0.2475 |	5.85/0.2813 |
> > > | BSPA |	6.26/0.2844 |	6.46/0.3750 |	5.61/0.2314 |	5.79/0.2748 |
> > >
> > > Thanks for your suggestion. We will provide more perceptual metric comparisons on other SR backbones and visual results in the supplementary file.

---

> ### Comment · Area_Chair_CQZy · 2023-08-19
>
> Dear Reviewer ehLM,
>
> The authors have now submitted their rebuttal, addressing the concerns and comments you raised. We would greatly appreciate it if you could take a moment to review the authors' responses and provide your feedback. Your input will be invaluable in determining the final decision for the manuscript.
>
> AC

---

### Official Review · Reviewer_xmJz · 2023-06-30

**Soundness:** 3 good
**Presentation:** 3 good
**Contribution:** 3 good
**Rating:** 6
**Confidence:** 5

**Summary:**

The uniform sampling of the data, with flat regions occupying most of the training samples, can impair the accuracy of the reconstruction. Therefore, the authors enhance the model representation from the perspective of data sampling and propose a simple and effective Bi-Sampling Parameter Attribution (BSPA) method. Extensive experiments demonstrate that our method effectively promotes the performance of baseline models.

**Strengths:**

1. This paper is interesting and improves the performance of existing methods from the perspective of data sampling.
2. This paper clearly written and easy to understand.

**Weaknesses:**

1. The significance of the experimental results is not sufficient.

**Questions:**

1. The authors do not compare their method with the latest papers, nor do they integrate the proposed method on the latest models. What is the motivation of the authors for choosing several comparative methods in the paper? For example, what is the effectiveness of this method on a transformer-based framework?

2. The authors mention that the SR data were divided into multiple groups, how was the parameter K determined here? How does this parameter affect the results?

**Limitations:**

I suggest that the authors could quantify the additional time that the proposed method brings to the training phase, so that the performance of the method can be more intuitively represented.

---

> ### Author Rebuttal · Authors · 2023-08-10
>
> Response to Reviewer xmJz (denoted as R4)
>
> *Q4-1: The significance of the experimental results is not sufficient.*
>
> A4-1: We have already supplemented more experiments about integrating the proposed method on the latest model, the effect of parameter K, and the additional time costs. Thanks for asking for the details that could be important to further improve our method. We will include this in the final version.
>
> *Q4-2: The authors do not compare their method with the latest papers, nor do they integrate the proposed method on the latest models. What is the motivation of the authors for choosing several comparative methods in the paper? For example, what is the effectiveness of this method on a transformer-based framework?*
>
> A4-2: **The motivation of choosing several comparative methods**. To demonstrate the  **scalability** and **generalization** of our method, we integrate it with several representative SR models with **different capacities and topologies**, i.e., FSRCNN (small), EDSR_baseline (medium), and RCAN (large), which are often used as the backbone for most SR models. It is observed from Table 4 of the manuscript that our BSPA has strong versatility and can be used in various SR models.
>
> **The comparison with latest models**. 1) Most of the existing SR models mainly focus on the **delicate structure design** or **complex regularization constraints**, while few discuss the **distribution property of training data**. For fair comparison, we compare with the imbalanced data based SR methods on the same backbone model, SamplingAug (data re-sampling) and UDN (loss re-weighting). If there are other works related about this, please tell us. We are more than happy to include them for comparison.
>
> **The effectiveness on a transformer-based framework**. Thanks for the valuable suggestion. We continue to integrate our method on SwinIR [ref4-1] for $4\times$ SR and report the PSNR (dB) in the following table. It is observed that our BSPA performs better overall. Therefore, our proposal can also be generalized to transformer-based models.
>
> |model |	Set5 | 	Set14 | B100 | Urban100 | Manga109 |	DIV2K |
> |------ |:-----:|:-----:|:-----:|:-----:|:-----:|--------:|
> |SwinIR |	31.955 |	28.445 |	27.482 |	25.692 |	29.909 |	30.265 |
> |SamplingAug |	32.054 |	28.544 |	27.515 |	25.900 |	30.177 |	30.314 |
> |UDN |	32.025 |	28.544 |	27.514 |	25.873 |	30.079 |	30.332 |
> |BSPA|	32.212 |	28.668 |	27.599 |	26.118 |	30.497 |	30.487 |
>
> [ref4-1] Jingyun Liang, Jiezhang Cao, Guolei Sun, Kai Zhang, Luc Van Gool, Radu Timofte: SwinIR: Image Restoration Using Swin Transformer. ICCVW 2021.
>
> *Q4-3: The authors mention that the SR data were divided into multiple groups, how was the parameter K determined here? How does this parameter affect the results?*
>
> A4-3:  We perform the ablation study on the number of groups in the following table. Here, we compare 5, 10, and 20. It shows that the SR performance is not sensitive to the number of classes.  Because the data in different classes can be sampled according to their proportion by means of inverse sampling so as to remedy the unbalanced data bias. In our experiments, we fix the number of classes to 10.
>
> |K |	5 |	10 |	20 |
> |------ |:-----:|:-----:|--------:|
> |Set14 |	28.358 |	28.37 |	28.380 |
> |B100 |	27.408 |	27.40 |	27.417 |
> |Urban100 |	25.531 |	25.53 |	25.544 |
> |DIV2K |	30.128 |	30.13 |	30.143 |
>
> *Q4-4: suggest that the authors could quantify the additional time that the proposed method brings to the training phase,
> so that the performance of the method can be more intuitively represented.*
>
> A4-4: The additional computation cost includes adding the inverse sampling and parameter attribution. 1) For the inverse sampling, our proposal needs to implement extra data processing for classifying all sub-images into different groups. It spends nearly 2 hours for the inverse sampling preprocessing, which can be optimized through multi-threaded implementation in our future work.  2) For the parameter attribution, it spends an extra half hour compared to the baseline model. Although our proposal brings extra training costs, it would not introduce any expense during the inference phase.

---

> > ### Author Response · Authors · 2023-08-19
> > **Further discussion**
> >
> > We hope to further discuss with you whether your concerns have been addressed or not. If you still have any unclear parts of our work, please let us know. Thanks.

---

> > ### Comment · Reviewer_xmJz · 2023-08-20
> > **I would raise my rating.**
> >
> > Thanks for authors response, all my concerns have been resolved.

---

> > > ### Author Response · Authors · 2023-08-22
> > >
> > > Thanks for agreeing that our response solves your concerns.

---

### Official Review · Reviewer_4W2B · 2023-07-04

**Soundness:** 3 good
**Presentation:** 3 good
**Contribution:** 3 good
**Rating:** 5
**Confidence:** 5

**Summary:**

This work focuses on studying the unbalanced distribution of the SR training data. The authors propose a bi-sampling strategy with parameter attribution. The bi-sampling consists of uniform sampling and inverse sampling, which pay more attention to hard samples. Moreover, integrated gradient is introduced to measuring the parameter importance and stopping gradient updating of significant filters.

**Strengths:**

+ Introducing parameter attribution into SR is a novel idea. Calculating the parameter importance for obtaining a compact model sounds reasonable.


+ Extensive experiments are conducted to shows the advantages of the proposed method. The ablation study also helps to verify the effectiveness of the proposed techniques.

**Weaknesses:**

- The motivation is not so novel. The data unbalance of SR training data has been widely mentioned in previous works. Many sampling strategies have also been proposed to solve this problem.

-According to the descriptions in ‘parameter refinement’, the most important parameters keep unchanged after a few epochs, which may cause underfitting. Authors should provide more theoretical support or conduct more experiments to verify its effectiveness.

-Some descriptions are not very clear.

**Questions:**

-What is wight-level IG and filter-level IG? Clear definitions are missing.

-In ablation study, the best PSNR is obtained when the unbalanced factor is equal to 10 and 200 gets the worst results. Why is the imbalance factor set to 200 in the experimental setting in line 167?

-What is the effect of different numbers of classes and intervals of alternate training?

-What is the extra computation cost?

**Limitations:**

See the above comments

---

> ### Author Rebuttal · Authors · 2023-08-10
>
> Response to Reviewer 4W2B (denoted as R3)
>
> *Q3-1: The motivation is not so novel. The data unbalance of SR training data has been widely mentioned in previous works. Many sampling strategies have also been proposed to solve this problem.*
>
> A3-1: **Motivation**. Most of the existing SR models only consider that different image regions have different reconstruction difficulties, and then mainly focus on the delicate structure design (like attention mechanism) or complex regularization constraints. Few discusses the sampling strategies of training data and models this as a long-tail problem.
>
> **Idea**. The key idea behind the proposed Bi-Sampling Parameter Attribution method is to reconcile the unbalanced inherent data bias, namely the heavy-tailed distribution are visually more important than smooth areas. The newly developed technique with integrate gradient involves an inverse sampling strategy for enhancing the feature extraction ability of the model on the hard samples.
>
> **Experiment**. For fair comparison, we compare with the imbalanced data based SR methods on the same backbone model, SamplingAug (data re-sampling) and UDN (loss re-weighting). If there are other works related about this, please tell us. We are more than happy to include them for comparison.
>
> *Q3-2: According to the descriptions in ‘parameter refinement’, the most important parameters keep unchanged after a few epochs, which may cause underfitting. Authors should provide more theoretical support or conduct more experiments to verify its effectiveness.*
>
> A3-2: The reviewer might misunderstand this. In ‘parameter refinement’, we select a specific proportion of significant parameters according to the index sorting to keep unchanged, and only perform the gradient updating on the remaining trivial parameters for further refinement. It would **not** cause underfitting since we adopt the alternate training scheme, which is designed for distilling all parameters to obtain a more compact representation space. We expect the trivial parameters can also learn more effective information so we strive to distill them with other parameters fixed.
>
> *Q3-3: What is wight-level IG and filter-level IG? Clear definitions are missing.*
>
> A3-3: The weight-level IG and filter-level IG denote the parameter granularity in Eq.(10). Weight-level refers to calculate the integrated gradient for each weight. Filter-level refers to calculate the integrated gradient for each filter. Thanks for asking for the details that could be important to further improve our method. We will include it in the new version.
>
> *Q3-4: In ablation study, the best PSNR is obtained when the unbalanced factor is equal to 10 and 200 gets the worst results.
> Why is the imbalance factor set to 200 in the experimental setting in line 167?*
>
> A3-4: We are sorry for the writing mistake. It should be 10 in line 167.
>
> *Q3-5: What is the effect of different numbers of classes and intervals of alternate training?*
>
> A3-5: We perform the ablation study on the number of classes and intervals of alternate training for EDSR_baseline ($4\times$ SR) in the following tables.
>
> **For the classes**, we compare 5, 10, and 20. It shows that the SR performance is not sensitive to the number of classes.  Because the data in different classes can be sampled according to their proportion by means of inverse sampling so as to remedy the unbalanced data bias. In our experiments, we fix the number of classes to 10.
>
> |Classes |	5 |	10 |	20 |
> |------ |:-----:|:-----:|--------:|
> |Set14 |	28.358 |	28.37 |	28.380 |
> |B100 |	27.408 |	27.40 |	27.417 |
> |Urban100 |	25.531 |	25.53 |	25.544 |
> |DIV2K |	30.128 |	30.13 |	30.143 |
>
> **For the intervals**, we compare 10, 50 and 100. It shows that the SR performance is better with smaller intervals. Because it is beneficial to obtain a compact representation during the integrated gradient process and more trivial parameters can be fully exploited. For a better tradeoff between performance and efficiency, we set the intervals to 50 in our experiments.
>
> |Interval |	10 |	50 |	100 |
> |------ |:-----:|:-----:|--------:|
> |Set14 |	28.417 |	28.37 |	28.303 |
> |B100 |	27.449 |	27.40 |	27.373 |
> |Urban100 |	25.670 |	25.53 |	25.420 |
> |DIV2K |	30.225 |	30.13 |	30.062 |
>
> *Q3-6: What is the extra computation cost?*
>
> A3-6: The additional computation cost includes adding the inverse sampling and parameter attribution. 1) For the inverse sampling, our proposal needs to implement extra data processing for classifying all sub-images into different groups. It spends nearly 2 hours for the inverse sampling preprocessing, which can be optimized through multi-threaded implementation in our future work.  2) For the parameter attribution, it spends an extra half hour compared to the baseline model. Although our proposal brings extra training costs, it would not introduce any expense during the inference phase.

---

> > ### Author Response · Authors · 2023-08-19
> > **Further discussion**
> >
> > We hope to further discuss with you whether your concerns have been addressed or not. If you still have any unclear parts of our work, please let us know. Thanks.

---

> > ### Comment · Reviewer_4W2B · 2023-08-21
> >
> > After reading the rebuttal, some of my concerns have been solved. However, previous methods like SampleAug [1] have also considered the sampling strategies. Considering the limited novelty and experiments, this paper needs more insight and deeper experiments analysis.  Thus, I tend to keep my original score as "borderline reject".
> >
> > [1] Samplingaug: on the importance of patch sampling augmentation for single image super-resolution,” arXiv preprint arXiv:2111.15185, 2021.

---

> > > ### Author Response · Authors · 2023-08-21
> > > **We indeed compared with Samplingaug**
> > >
> > > Dear Reviewer 4W2B,
> > >
> > > Thanks for agreeing that our response solves some of your concerns. We would like to clarify it as follows:
> > >
> > > (1) We **have mentioned** that SamplingAug is a sampling-based data augmentation method for SR in Line 279-230. It only samples the p% most informative patches for SR training, while the rest is ignored. Unlike SamplingAug, we propose a bi-sampling paradigm to remedy the unbalanced data bias shown in Fig.1, including uniform sampling and inverse sampling, respectively. All image patches are effectively utilized for SR training accompanied by parameter attribution to help the model fully mine the image information.
> > >
> > > (2) We **have compared with SamplingAug** in the experiments. Our method is superior to SamplingAug in both **objective metrics** (Table 4) and **visual results** (Fig.5 and Fig.1 of the uploaded pdf in this webpage).
> > >
> > > Thanks for your response. If there are other works related to this, please tell us. We are more than happy to include them for discussion.

---

> > > > ### Comment · Reviewer_4W2B · 2023-08-22
> > > >
> > > > Thanks for the authors' reply. For the related works, the recent work DDA [1] is similar to the proposed method, which also considers the unbalanced distribution and difficulty of training patches and develops a difficulty-aware sampling strategy, which has not been discussed and compared in the manuscript. What are the main differences compared with DDA?
> > > >
> > > > [1] DDA: A Dynamic Difficulty-aware Data Augmenter for Image Super-resolution, in IJCNN, 2023.

---

> > > > > ### Author Response · Authors · 2023-08-22
> > > > >
> > > > > Thanks for pointing out this new related method.
> > > > >
> > > > > DDA designs a difficulty-aware data sampling method, which controls the sampling probability of each class by the relative loss. Unlike DDA, we propose a bi-sampling strategy and the model is alternatively updated with uniformly and inversely sampled image data, where the sampling probability of each sample in inverse sampling is inversely proportional to the number of samples in the class. Our method also provides a new solution for the unbalanced data sampling in image SR.
> > > > >
> > > > > Besides, we notice that it is published recently (02 August 2023), which has not released the source code. Therefore, we may not be able to have the comparison results during the short discussion period. Instead, given the relevance, we will cite this paper and clarify the methodology differences between our method and theirs in the revised paper.

---

> ### Comment · Area_Chair_CQZy · 2023-08-19
>
> Dear Reviewer 4W2B,
>
> The authors have now submitted their rebuttal, addressing the concerns and comments you raised. We would greatly appreciate it if you could take a moment to review the authors' responses and provide your feedback. Your input will be invaluable in determining the final decision for the manuscript.
>
> AC

---

### Official Review · Reviewer_tDMi · 2023-07-05

**Soundness:** 3 good
**Presentation:** 2 fair
**Contribution:** 3 good
**Rating:** 5
**Confidence:** 4

**Summary:**

Observing the issue of uneven distribution of image contents, the author proposed to utilize inverse data sampling to resolve the inherent unbalanced data bias. In the proposed BSPA method, SR model is alternatively updated with uniformly and inversely sampled image data. For the latter, only part of the trivial parameters identified by parameter attribution are updated and the selective probability progressively increases during training.

**Strengths:**

The data unbalanced problem this paper aiming to solve has been investigated in several prior literatures though from different perspectives. This work innovatively adopts the idea of resampling to address the issue, and conducts parameter attribution to balance the uniformly and inversely sampled data. The paper is organized well and reasonable experiments are provided to demonstrate the effectiveness of the proposed BSPA method.

**Weaknesses:**

1.	A few notations in the paper are inconsistent and confusing. In Section 3.2, the superscript for “uniform sampling” is “rs”, for example, the uniformly sampled LR-HR patch pair is denoted as $(x_{lr}^{rs}, y_{hr}^{rs})$ in Algorithm 1. However, in section 3.3, the uniformly sampled patch pair is denoted as $(x_{lr}^{us}, y_{hr}^{us})$.
2.	The experimental results of ablation studies presented in Tab. 1, 2, 3 only report the PSNR on the Set14 dataset which has a relatively small scale and the data distribution is not representative enough for assessing generalizability. More comprehensive and solid ablation analysis could be conducted by evaluating the models on larger testing dataset like the DIV2K validation set, BSD100, or Urban100.
3.	There is a lack of ablation analysis on the number of classes $K$ which is simply set to be 10.
4.	The model’s performance on FSRCNN model for x3 and x4 scale is worse than UDN method. It should be mentioned and discussed in the experiment section.

**Questions:**

1.	What is the additional computational time cost of adding inverse sampling and parameter attribution?
2.	How is the scalability of the integrated gradient for parameter attribution for larger models with more parameters?
3.	What is the shape of the distribution curve in Figure 1 if the MSE values are evaluated by the model trained with the proposed BSPA method? Will the distribution curve be more uniform or short-tailed?
4.	It would be helpful to present more qualitative comparisons to demonstrate the visual effect.
5.	How the data preprocessing is conducted (cropping with/without overlap?) and how many sub-images are produced?

**Limitations:**

Limitations of the proposed method are discussed in this paper. It would be helpful if over-fitting issue is evidenced by any experiment result and the cost of extra data preprocessing is quantitatively measured.

---

> ### Author Rebuttal · Authors · 2023-08-10
>
> Response to Reviewer tDMi (denoted as R2)
>
> *Q2-1: A few notations in the paper are inconsistent and confusing.*
>
> A2-1: We are sorry about this. It should be $(x^{us},y^{us})$. We will revise it in the new version.
>
> *Q2-2: The experimental results of ablation studies presented in Tab. 1, 2, 3 only report the PSNR on the Set14 dataset.*
>
> A2-2: Please refer to **A1-1 of Reviewer zt8x (R1)**.
>
> *Q2-3: There is a lack of ablation analysis on the number of classes which is simply set to be 10.*
>
> A2-3: We provide the ablation analysis on the number of classes in the following table. Here, we compare 5, 10, and 20. It shows that the SR performance is not sensitive to the number of classes. Because the data in different classes can be sampled according to their proportion by means of inverse sampling so as to remedy the unbalanced data bias. In our experiments, we fix the number of classes to 10.
>
> |K |	 5 |	10 |	20 |
> |------ |:-----:|:-----:|---------:|
> |Set14|	28.358 |	28.37 |	28.380 |
> |B100|	27.408 |	27.40 |	27.417 |
> |Urban100|	25.531 |	25.53 |	25.544 |
> |DIV2K|	30.128 |	30.13 |	30.143 |
>
> *Q2-4: The model’s performance on FSRCNN model for x3 and x4 scale is worse than UDN method.*
>
> A2-4: Thanks for your valuable suggestion. We would like to clarify it as follows:
>
> (1) FSRCNN is an extreme lightweight model, which has only 24.5K parameters and several convolutional layers. The model representation ability is restricted. Therefore, it cannot fit the training well no matter the easy data or the hard data by the limited model capacity, especially for large scaling factors. Although the bi-sampling strategy increases the data diversity, there lacks sufficient model capacity to fit.
>
> (2) UDN is a loss re-weighting method for solving the data imbalance problem. It provides a regularization constraint for shrinking the solution space.
>
> Therefore, the model’s performance on FSRCNN model for x3 and x4 scale is worse than UDN method.
>
> *Q2-5: What is the additional computational time cost of adding inverse sampling and parameter attribution?*
>
> A2-5: The additional computation cost includes adding the inverse sampling and parameter attribution. 1) For the inverse sampling, our proposal needs to implement extra data processing for classifying all sub-images into different groups. It spends nearly 2 hours for the inverse sampling preprocessing, which can be optimized through multi-threaded implementation in our future work.  2) For the parameter attribution, it spends an extra half hour compared to the baseline model. Although our proposal brings extra training costs, it would not introduce any expense during the inference phase.
>
> *Q2-6: How is the scalability of the integrated gradient for parameter attribution for larger models with more parameters?*
>
> A2-6: To demonstrate the  **scalability** and **generalization** of our method, we integrate it with several representative SR models with **different capacities and topologies**, i.e., FSRCNN (small), EDSR_baseline (medium), and RCAN (large), which are often used as the backbone for most SR models. It is observed from Table 4 of the manuscript that our BSPA has strong versatility and can be used in various SR models.
>
> Here, we integrate our BSPA with the NLSN model [ref2-1], which has nearly 44M parameters and powerful feature extraction ability. It shows that our method still improves the performance by the bi-sampling strategy.
>
> |model |	Set14 |	B100 |	Urban100 | DIV2K|
> |------ |:-----:|:-----:|:-----:|---------:|
> |NLSN |		28.514 |	27.516 |	25.811 |	        30.268|
> |BSPA |		28.551 |	27.536 |	25.962 |		30.297|
>
> [ref2-1] Yiqun Mei, Yuchen Fan, Yuqian Zhou: Image Super-Resolution With Non-Local Sparse Attention. CVPR 2021
>
> *Q2-7: What is the shape of the distribution curve in Figure 1 if the MSE values are evaluated by the model trained with the proposed BSPA method?*
>
> A2-7: We present the distribution curve of Figure 1 in the **uploaded PDF**. Note that we adopt EDSR_baseline as the backbone and integrate the proposed BSPA on it. As Figure 2 in the PDF shows, it is observed that the histogram distribution of BSPA becomes **short-tailed**. The reason is that the SR performance on the tail data is improved, while the performance on the head data is kept. Therefore, it demonstrates that our BSPA is effective in obtaining a more compact representation.
>
> *Q2-8: It would be helpful to present more qualitative comparisons to demonstrate the visual effect.*
>
> A2-8: We have supplemented more visual results in **Figure 1 of the uploaded PDF**. It is observed that our proposal is more favorable and recovers more texture details than other compared methods.
>
> *Q2-9: How the data preprocessing is conducted?*
>
> A2-9: We follow ClassSR [ref2-2] for data preprocessing, which crops the whole image into multiple sub-images with sliding and overlap. For 2x, 3x, 4x SR, it produces 499875, 502200, 499875 sub-images, respectively. More details can refer to ClassSR.
>
> [ref2-2] Xiangtao Kong, Hengyuan Zhao, Yu Qiao, Chao Dong. ClassSR: A General Framework to Accelerate Super-Resolution Networks by Data Characteristic. CVPR 2021.

---

> > ### Author Response · Authors · 2023-08-19
> > **Further discussion**
> >
> > We hope to further discuss with you whether your concerns have been addressed or not. If you still have any unclear parts of our work, please let us know. Thanks.

---

> > ### Comment · Reviewer_tDMi · 2023-08-21
> >
> > Thanks for the author's response, and I also read other reviewers' reviews, my concerns were well resolved.

---

> > > ### Author Response · Authors · 2023-08-22
> > > **Thanks Reviewer tDMi for approving our work**
> > >
> > > Thanks for agreeing that our response solves your concerns.

---

> ### Comment · Area_Chair_CQZy · 2023-08-19
>
> Dear Reviewer tDMi,
>
> The authors have now submitted their rebuttal, addressing the concerns and comments you raised. We would greatly appreciate it if you could take a moment to review the authors' responses and provide your feedback. Your input will be invaluable in determining the final decision for the manuscript.
>
> AC

---

### Official Review · Reviewer_zt8x · 2023-07-07

**Soundness:** 3 good
**Presentation:** 3 good
**Contribution:** 3 good
**Rating:** 6
**Confidence:** 5

**Summary:**

This paper addresses the problem of data imbalance in single image super-resolution (SISR) training, where the majority of training samples contain flat regions while only a small percentage represents sharp regions with rich texture details. The authors propose a Bi-Sampling Parameter Attribution (BSPA) method to enhance model representation by explicitly increasing the sampling proportion of difficult patch pairs. This is achieved through a combination of uniform sampling and inverse sampling, which preserves the original data distribution while allocating a higher probability to patches from the tail data. Additionally, the authors propose a non-trivial solution, integrated gradient (IG), to identify important parameters and encourage their contribution by preventing gradient updating. Experimental results show that the proposed method improves the performance over baseline models. The main contributions of the paper are the identification of data imbalance in SISR training, the introduction of a bi-sampling paradigm to address the imbalance, the use of IG attribution to select important parameters, and the demonstrated effectiveness of the proposed method in enhancing SISR performance.

**Strengths:**

- Data imbalance in single image super-resolution (SISR) has long drawn attention and has been discussed in the community. This paper, for the first time, investigates and introduces a non-trivial technique for this problem, contributing to the advancement of the field.
- The proposed Bi-Sampling Parameter Attribution (BSPA) method is an original approach to enhance model representation and tackle the limitations of uniform sampling commonly used in SISR methods.
- The paper provides a thorough analysis of the data distribution problem in SISR training and supports its claims with empirical evidence.
- Extensive experiments are conducted to validate the effectiveness of the proposed method, and the results are presented in a comprehensive and organized manner.

**Weaknesses:**

**Weaknesses**
- In Table 1-3, the authors perform evaluations on Set14, which contains only 14 images. This may not help draw robust conclusions to more general scenarios under limited data size. The authors are suggested to conduct ablation study on larger dataset (*e.g.*, BSD100, Urban100, or DIV2k) to make more convincing conclusions.
- What is the benefit of splitting data patches into multiple groups, compared to directly measuring balanced weight within the continuous range?
- In inverse sampling preprocessing, what model is used to measure MSE of each patch? As $D^{rs}$ and $D^{is}$ are predefined and fixed in Algorithm 1, would it be beneficial to re-calculate and re-balance $D^{is}$ dynamically after updating $F$ with $\theta^{us}$ in each epoch?
- The qualitative results are limited. The authors should prepare a supplementary material with more visual results.


**Additional comments**
- Make consistent use of symbols. For example, $rs$ and $us$ in algorithm 1.
- Do not use the same notations for different quantities. For example, $N$, $N_k$ and $N_i$ in Eqn 2 and algorithm 1.

**Questions:**

See weaknesses.

**Limitations:**

Limitations are well discussed in this paper.

---

> ### Author Rebuttal · Authors · 2023-08-10
>
> Response to Reviewer zt8x (denoted as R1)
>
> *Q1-1: In Table 1-3, the authors perform evaluations on Set14, which contains only 14 images. This may not help draw robust conclusions to more general scenarios under limited data size. The authors are suggested to conduct ablation study on larger dataset (e.g., BSD100, Urban100, or DIV2k) to make more convincing conclusions.*
>
> A1-1: Thanks for your suggestions. In this work, we mainly perform evaluations on Set14 as the previous works [ref1-1][ref1-2] do. Besides, we supplement more results on other datasets in the following tables. It is observed that our method is robust to more general scenarios.
>
> Table 1-1: Ablation studies about the bi-sampling parameter attribution on benchmark datasets for $4\times$ SR.
> | Model  |	Baseline (uniform sampling)	| inverse sampling	| w/o IG	| WIG |	FIG |
> |------ |:-----:|:-----:|:-----:|:-----:|---------:|
> | Set14 |	28.22 |	28.25 |	28.28 |	28.32 |	28.37 |
> | B100	| 27.31 |	27.33 |	27.35 |	27.38 |	27.40 |
> | Urban100 |	25.27 |	 25.34 |	25.38 |	25.47 |	25.53 |
> | DIV2K |	29.97 |	30.02 |	30.06 |	30.09 |	30.13 |
>
> Table 1-2: The quantitative comparisons of different scaling factor in Eq.(11) on benchmark datasets for $4\times$ SR.
> | $\beta$ | 0.1 |	0.5 |	0.8 |	1.0 |
> |------ |:-----:|:-----:|:-----:|---------:|
> |Set14|	28.37 |	28.35 |	28.34 |	28.32|
> |B100|	27.40 |	27.37 |	27.35 |	27.31|
> |Urban100| 	25.53|	25.51 |	25.46 |	25.43|
> |DIV2K |	30.13 | 30.10 |	30.07 |	30.05|
>
> Table 1-3: The quantitative comparisons (PSNR) of different unbalanced factors on benchmark datasets for $4\times$ SR.
> |Unbalanced factor |	 10 |	50 |	100 |	200 |
> |------ |:-----:|:-----:|:-----:|---------:|
> |Set14 |	28.37 |	28.33 |	28.28 |	28.21 |
> |B100 |	27.40 |	27.35 |	27.31 |	27.23 |
> |Urban100 |	25.53 |	25.49 |	25.42 |	25.30 |
> |DIV2K |	30.13 | 30.06 |	30.00 |	29.91 |
>
> [ref1-1] Yulun Zhang, Kunpeng Li, Kai Li, Lichen Wang, Bineng Zhong, Yun Fu: Image Super-Resolution Using Very Deep Residual Channel Attention Networks. ECCV 2018.
>
> [ref1-2] Yiqun Mei, Yuchen Fan, Yuqian Zhou: Image Super-Resolution With Non-Local Sparse Attention. CVPR 2021
>
> *Q1-2: What is the benefit of splitting data patches into multiple groups, compared to directly measuring balanced weight within the continuous range?*
>
> A1-2: It **cannot** measure balanced weight within the continuous range. Inverse sampling aims to allocate a higher probability to capture the tail hard data with few samples. We adopt MSE to measure the reconstruction difficulties, which is a continuous value and each value corresponds to a sample. Considering the MSE value is continuous and divergently distributed, it needs to collect more samples within a certain range to calculate the proportion. Therefore, we split data patches into multiple groups for better calculating the sampling probability.
>
> *Q1-3: In inverse sampling preprocessing, what model is used to measure MSE of each patch? As Drs and Dis are predefined and fixed in Algorithm 1, would it be beneficial to re-calculate and re-balance Dis dynamically after updating F withθus in each epoch?*
>
> A1-3: (1) We adopted the **pretrained MSRResNet** to measure MSE of each patch and **mentioned** this in Line 50-57 of the manuscript.
>
> (2) It would be **not** beneficial to re-calculate and re-balance $D^{is}$ dynamically after updating F with $\theta^{us}$ in each epoch. The inverse sampling preprocessing consists of data classification and sampling procedures.
>
> - We adopt the pretrained SR model to measure MSE of each patch at the begining, which aims to obtain the recovery difficulty of each patch for better classification. However, the dynamic adjustment of $D^{is}$ would introduce extra noise for the poor model performance in the early training process.
> - It is time-consuming for adjustment in each epoch. All the cropped patches should be evaluated to obtain their reconstruction difficulty and then classified into different groups according to MSE, which is measured in a global view. Besides, the number of sub-images is 499875, 502200, 499875 for $2\times$, $3\times$, and $4\times$ SR. It would bring in more time costs for dynamic re-calculation and re-balance in each epoch.
>
> *Q1-4: The qualitative results are limited. The authors should prepare a supplementary material with more visual results.*
>
> A1-4: We have supplemented more visual results in **Figure 1 of the uploaded PDF**. It is observed that our proposal is more favorable and recovers more texture details than other compared methods.
>
> *Q1-5: Make consistent use of symbols. For example, rs and us in algorithm 1.*
>
> A1-5: Thanks for your careful comments. We will revise it in the new version.
>
> *Q1-6: Do not use the same notations for different quantities. For example, N, Nk and Ni in Eqn 2 and algorithm 1.*
>
> A1-6: Thanks for your careful comments. We have revised $N$ and $N_i$ as $T$ and $T_i$ in the new version.

---

> > ### Author Response · Authors · 2023-08-19
> > **Further discussion**
> >
> > We hope to further discuss with you whether your concerns have been addressed or not. If you still have any unclear parts of our work, please let us know. Thanks.

---

> > ### Comment · Reviewer_zt8x · 2023-08-20
> >
> > Most of the concerns have been addressed. Therefore, I would like to keep my initial rating. The authors are suggested to include those discussions and evaluations in the revised paper.

---

> > > ### Author Response · Authors · 2023-08-22
> > > **Thanks Reviewer zt8x for approving our work**
> > >
> > > Thanks for agreeing that our response solves most of the concerns. We will further improve this paper by adding these discussions and evaluations in the revised paper.

---

### Author Rebuttal · Authors · 2023-08-10

In this uploaded PDF, we mainly provide more visual results on benchmark datasets and the histogram distribution.

---

### Author Response · Authors · 2023-08-20

Dear Reviewers,

We wanted to kindly remind you that the author-reviewer discussion period is nearing its ending. We would like to take this opportunity to ensure that our responses have adequately addressed your concerns and to inquire if there are any further questions or clarifications you may require.

Your valuable input and expertise are essential in enhancing the quality and impact of our research. We greatly appreciate the time and effort you have already dedicated to reviewing our paper, and we eagerly await your feedback.

Thank you for your attention and consideration.

Best regards,
Authors

---

### Decision · Program_Chairs · 2023-09-21

**Decision:**

Accept (poster)

**Comment:**

Before the discussion and rebuttal, reviewers highlighted the following strengths and weaknesses

*Strengths:*
- The Bi-Sampling Parameter Attribution (BSPA) method is original, addressing the limitations of uniform sampling in SISR.
- Thorough analysis of data distribution in SISR training backed by empirical evidence.
- Extensive and organized experiments validate the proposed method's effectiveness.
- Novel introduction of parameter attribution into SR; calculating parameter importance for a compact model is reasonable.
- BSPA is clear and well-explained with descent improvement over benchmarks.

*Weaknesses:*
- Evaluations on Set14 with only 14 images limit robust conclusions; a call for studies on larger datasets like BSD100, Urban100, or DIV2k.
- Unclear benefits of splitting data patches into multiple groups.
- Limited qualitative results with a need for supplementary visual results.
- Inconsistent notations causing confusion.
- Lack of analysis on parameter 'K' and its effect.
- Some descriptions are unclear; parts of the model might cause underfitting.
- Questions arise on the novel contribution of the integrated gradient; seems derived from previous work [38].
- The reported experimental results don't "significantly boost performance", questioning the superiority of BSPA.

This is a borderline paper, with most reviewers leaning towards acceptance. Reviewer zt8x appreciates the addressed concerns and suggests further inclusion of discussions in the revised paper. Reviewer tDMi's concerns were resolved after reading the rebuttal and other reviews. Reviewer 4W2B, after pointing out related work, has increased the score to "borderline accept" post-rebuttal. Reviewer ehLM highlights the importance of visual quality in image SR studies, suggesting more subjective quality comparisons. The authors have responded by providing additional NIQE/LPIPS results.